# Eliminating chronic myeloid leukemia stem cells by IRAK1/4 inhibitors

Yosuke Tanaka [1✉], Reina Takeda [1], Tsuyoshi Fukushima[1], Keiko Mikami[1], Shun Tsuchiya[2], Moe Tamura[1], Keito Adachi[1], Terumasa Umemoto [3], Shuhei Asada [1], Naoki Watanabe[2], Soji Morishita [4], Misa Imai[4], Masayoshi Nagata[5], Marito Araki [4], Hitoshi Takizawa [3], Tomofusa Fukuyama [1], Chrystelle Lamagna[6], Esteban S. Masuda[6], Ryoji Ito [7], Susumu Goyama[8], Norio Komatsu[2], Tomoiku Takaku[2] & Toshio Kitamura[1✉]

Leukemia stem cells (LSCs) in chronic myeloid leukemia (CML) are quiescent, insensitive to BCR-ABL1 tyrosine kinase inhibitors (TKIs) and responsible for CML relapse. Therefore, eradicating quiescent CML LSCs is a major goal in CML therapy. Here, using a $G_0$ marker ($G_0M$), we narrow down CML LSCs as $G_0M$- and CD27- double positive cells among the conventional CML LSCs. Whole transcriptome analysis reveals NF-κB activation via inflammatory signals in imatinib-insensitive quiescent CML LSCs. Blocking NF-κB signals by inhibitors of interleukin-1 receptor-associated kinase 1/4 (IRAK1/4 inhibitors) together with imatinib eliminates mouse and human CML LSCs. Intriguingly, IRAK1/4 inhibitors attenuate PD-L1 expression on CML LSCs, and blocking PD-L1 together with imatinib also effectively eliminates CML LSCs in the presence of T cell immunity. Thus, IRAK1/4 inhibitors can eliminate CML LSCs through inhibiting NF-κB activity and reducing PD-L1 expression. Collectively, the combination of TKIs and IRAK1/4 inhibitors is an attractive strategy to achieve a radical cure of CML.

[1] Division of Cellular Therapy, The Institute of Medical Science, The University of Tokyo, Tokyo 108-8639, Japan. [2] Department of Hematology, Juntendo University Graduate School of Medicine, Tokyo 113-8421, Japan. [3] International Research Center for Medical Sciences, Kumamoto University, Kumamoto 860-0811, Japan. [4] Department of Transfusion Medicine and Stem Cell Regulation, Juntendo University Graduate School of Medicine, Tokyo 113-8421, Japan. [5] Department of Urology, Juntendo University Graduate School of Medicine, Tokyo 113-8421, Japan. [6] Rigel, South San Francisco, California 94080, USA. [7] Central Institute for Experimental Animals, Kanagawa 210-0821, Japan. [8] Division of Molecular Oncology Department of Computational Biology and Medical Sciences, Graduate School of Frontier Sciences, The University of Tokyo, Tokyo 108-8639, Japan. ✉email: ytims@ims.u-tokyo.ac.jp; kitamura@ims.u-tokyo.ac.jp

Chronic myeloid leukemia (CML) results from the transformation of hematopoietic stem cells (HSCs) by BCR-ABL1 fusion protein[1–3]. BCR-ABL1, which arises from the chromosomal translocation t(9;22), encodes a 210 kD chimeric protein (p210 BCR-ABL1) with constitutive tyrosine-kinase activity. BCR-ABL1 tyrosine kinase stimulates proliferation, activates pro-survival signaling pathways, inhibits apoptosis, and contributes to genomic instability to drive the disease[4–7]. The development of the first BCR-ABL1 tyrosine kinase inhibitor (TKI) imatinib has dramatically improved CML therapy and the survival of CML patients[8,9]. However, TKI treatment fails to eliminate CML leukemia stem cells (LSCs) even in patients reaching the stage of complete molecular responses, and the residual CML LSCs can be identified in a quiescent HSC fraction[10–14]. Even the second-generation TKIs nilotinib and dasatinib are not effective at eradicating quiescent CML LSCs[15,16]. CML LSCs share many features with normal HSCs, such as the quiescent state, localization in the hypoxia niche, and self-renewal ability[17–21]. Their quiescent state is one of the major reasons why CML LSCs are resistant to BCR-ABL1 TKIs[22,23]. This is probably because quiescent CML LSCs are not absolutely dependent on BCR-ABL activity for their survival[12,24]. In fact, the abrogation of the quiescent state of CML LSCs by Fbxw7 ablation increases their sensitivity to imatinib[25]. Moreover, the combination of histone deacetylase (HDAC) inhibitors, which induce apoptosis in quiescent cancer cell lines, with imatinib induces apoptosis in quiescent CML LSCs[26,27]. Thus, the ablation of the quiescence of CML LSCs is a key to enhancing their sensitivity to TKIs. On the other hand, imatinib partially contributes to their quiescent state[28]. Overall, how the quiescent state of CML LSCs is regulated in the presence of TKIs has yet to be elucidated.

NF-κB activation by BCR-ABL1 has been observed in culture and in a mouse CML-like model[29,30]. Inhibitors of NF-κB activation attenuate the lymphoid and myeloid leukemogenesis by BCR-ABL1 and decrease the number of CML LSCs in the mouse model[31]. NF-κB blockade by PS-1145, an IκB kinase (IKK) inhibitor, is effective at suppressing the growth of imatinib-resistant CML cell lines and bone marrow (BM) cells of imatinib-resistant CML-chronic phase (CML-CP) patient bone marrow (BM) cells in vitro[32]. These reports indicate that NF-κB activation is responsible for the TKI-resistance of CML LSCs.

Notably, CML LSCs express higher levels of IL-1 receptors (IL-1Rs) and are more sensitive to IL-1-induced NF-κB signaling than normal HSCs. Accordingly, the combination of a recombinant IL-1R antagonist (IL-1RA) with a TKI resulted in significantly greater inhibition of CML LSCs compared with the TKI alone[33]. Additionally, primitive CD34+CD38- CML-CP cells proliferate and activate NF-κB in response to IL-1β stimulation[34]. These reports indicate that the IL-1R-NF-κB signal axis is important for the proliferation of CML LSCs.

Although IL-1Rs share their downstream signaling pathways with Toll-like receptors (TLRs) except for TLR3, TLR signaling in CML has not been well examined. IRAK1/4, two serine/threonine kinases, are mediators of IL-1R/TLR-NF-κB signaling pathways. It was reported that the blockade of IRAK1/4 is effective at eradicating myelodysplastic syndrome (MDS) cells by inhibiting growth and inducing apoptosis through the deactivation of NF-κB[35]. The inhibition of IRAK1 is also effective at reducing the growth of acute myeloid leukemia (AML) cell lines and primary AML patient cells[36]. These studies indicate that IRAK1 is a key factor for the survival of myeloid leukemia cells. Therefore, the development of effective therapeutic approaches that target the IRAK1/4-NF-κB pathway with TKIs could be considered an attractive option for the eradication of CML LSCs.

The immune system plays an important role in the control of most malignancies including CML, and CML cells are susceptible to T cell immunity[37]. Among hematologic neoplasms, CML is sensitive to graft-versus-leukemia immunity[38]. The combination therapy of donor-derived CD8 T cells and imatinib results in prolonged leukemia-free survival of mice with BCR-ABL1-induced CML-like disease[39]. In relation to this, CML cells express programmed cell death ligand 1 (PD-L1), and a high expression of PD-L1 on human CD34+ CML cells is a negative prognostic factor[40,41]. Furthermore, PD-L1 is constitutively expressed on LSCs and leukemia progenitors in a mouse CML-like model. Finally, blocking the PD-1/PD-L1 interaction enhances the survival of CML mice in blast crisis[42]. These studies indicate that PD-L1 protects CML LSCs from T cell immunity. However, the efficacy of the combination of PD-1/PD-L1 blockade and TKIs on the eradication of CML LSCs has not been examined.

Previously we reported that G0 marker (G0M), a fusion protein between the fluorescent protein mVenus and p27K-, which is a p27 mutant lacking cyclin-dependent kinase (Cdk) inhibitory activity, is a useful tool for visualizing the quiescent states of conventional long-term HSCs, and the stemness of the cells was correlated with the intensity of G0M[43,44]. In the present study, we applied G0M to identify and visualize quiescent CML LSCs. Gene expression analysis revealed that the NF-κB signal is important for their survival in the presence of imatinib. The deactivation of NF-κB by an IRAK1/4 inhibitor with imatinib was extremely effective at eliminating CML LSCs compared with imatinib alone in a mouse CML-like model and a xenograft model with primary CML-CP patient samples. Moreover, we found that the IRAK1/4 inhibitor attenuated PD-L1 expression on CML LSCs in addition to having a proapoptotic function. These data indicate that the pharmacological inhibition of IRAK1/4 together with TKIs exerts a potent dual antileukemia effect partly through the assistance of antitumor immunity.

## Results

**Identification of CML LSCs by G0 marker and CD27.** We introduced G0 marker (G0M) to a mouse CML-like model to visualize quiescent CML LSCs. We retrovirally overexpressed BCR-ABL1 in BM cells from 5FU-treated G0M mice, in which Cre-inducible G0M in a *Rosa26* allele is regulated by Vav1-Cre[44], and then injected the cells into lethally irradiated wild-type recipient mice to develop mouse CML-like disease (Fig. 1a). CML derived from BM cells carrying G0M was developed in about 3 weeks, and imatinib treatment prolonged the survival of leukemic mice (Fig. 1b), ensuring that G0M did not affect CML development or imatinib treatment. To assess the expression of G0M in CML LSCs, we subdivided the conventional CML LSC fraction (CML LSK; BCR-ABL1+Lin−Sca1+cKit+) in the BM from leukemic mice by G0M and CD27, a marker for the CML stem/progenitor cell fraction[45]. The combination of CD27 and G0M split the CML LSK fraction into four factions: G0M+CD27+ (double positive: DP), G0M−CD27+ (single positive: SP), G0M+CD27−, and G0M−CD27− (double negative: DN) (Fig. 1c). To evaluate the LSC potential, we performed colony-forming assays and secondary transplantation assays on these fractions. The colony-forming ability was highly enriched in the DP fraction (Supplementary Fig. 1a). Kinetic analysis of the chimerism of BCR-ABL1+ cells (GFP+ cells) in peripheral blood (PB) revealed that, while the levels of chimerism gradually increased in the PB of mice transplanted with DP cells, engraftment of SP cells peaked at 25 days and then declined (Supplementary Fig. 1b). Chimerism of GFP+ myeloid cells in PB also increased in DP cell-engrafted mice, whereas that in SP cell-engrafted mice decreased and was undetectable at day 47 after the transplantation (Supplementary Fig. 1c). The secondary transplantation assay showed that mice

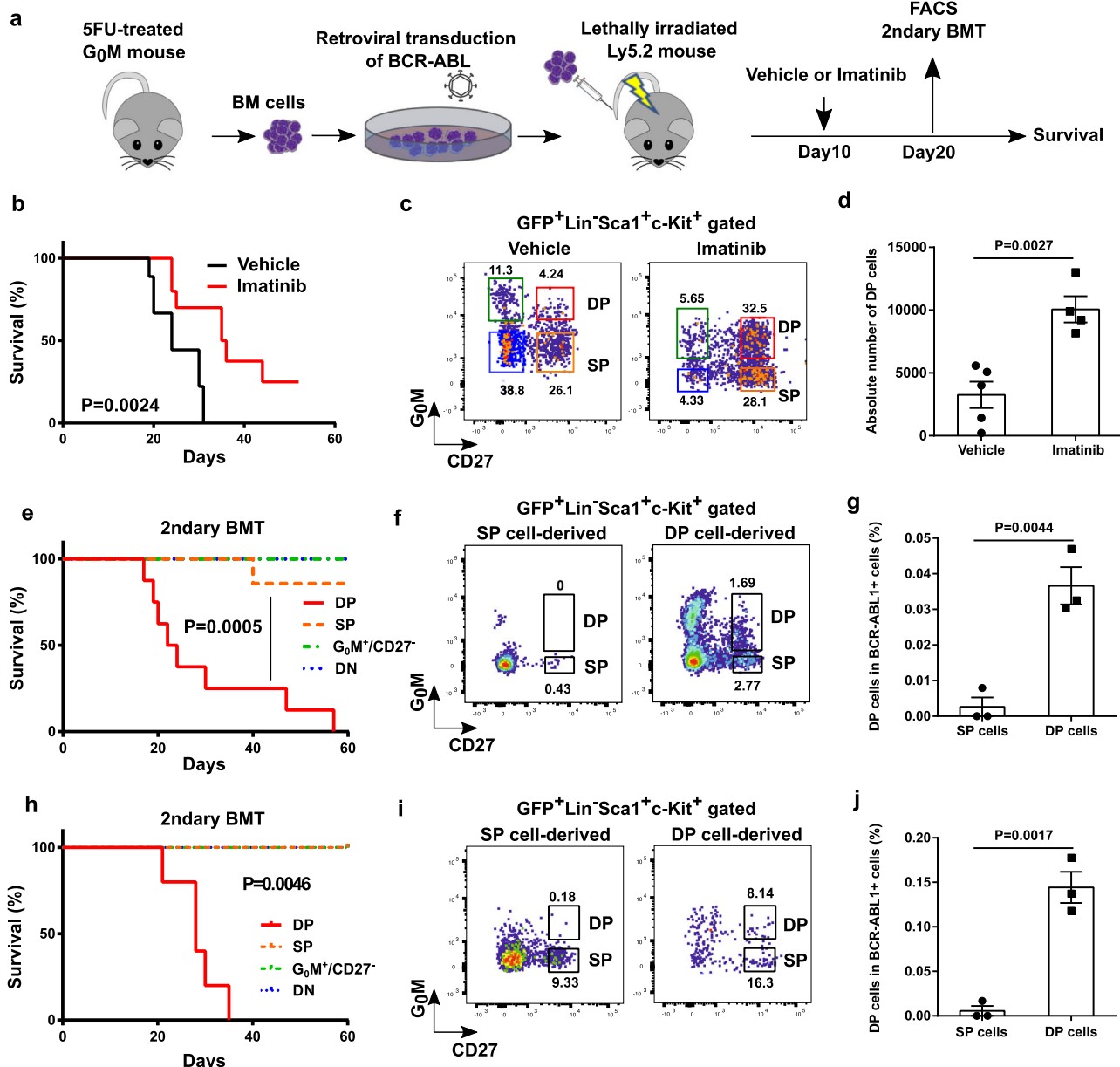

**Fig. 1 G$_0$M and CD27 identify quiescent CML LSCs. a** Experimental design. BMT: bone marrow transplantation. **b** Survival curves for mice treated with vehicle ($n = 9$) and imatinib ($n = 11$). **c** Representative FACS plots of BCR-ABL1+Lin-Sca1+c-Kit+ cells in the BM from mice treated with vehicle and imatinib. **d** Absolute number of LSCs (G$_0$M+CD27+ cells in FACS plots in **c**) in the BM from CML mice transplanted with SP cells ($n = 5$) and DP cells ($n = 4$). **e** Survival curves for recipient mice transplanted with the indicated populations from CML mice treated with vehicle. DP ($n = 8$), SP ($n = 8$), G$_0$M+CD27- ($n = 8$), and DN ($n = 7$). **f** Representative FACS plots of BCR-ABL1+Lin-Sca1+c-Kit+ cells in the BM from the recipient mice transplanted with DP and SP cells in **e**. **g** The percentage of LSCs (G$_0$M+CD27+ cells in FACS plots in F) in BCR-ABL1+ cells in the BM from CML mice transplanted with SP cells ($n = 3$) and DP cells ($n = 3$). **h** Survival curves for recipient mice transplanted with the indicated populations from CML mice treated with imatinib. DP ($n = 5$), SP ($n = 4$), G$_0$M+CD27- ($n = 3$), and DN ($n = 3$). **i** Representative FACS plots of BCR-ABL1+Lin-Sca1+c-Kit+ cells in the BM from the recipient mice transplanted with DP and SP cells in H. **j** The percentage of LSCs (G$_0$M+CD27+ cells in FACS plots in **i**) in BCR-ABL1+ cells in the BM from CML mice transplanted with SP cells ($n = 3$) and DP cells ($n = 3$). All treatments were continued during the observation period. Data are shown as the mean ± SEM. $P$ values are calculated by the log-rank test (**b, e, h**) or by two-tailed Student's $t$ test (**d, g, j**).

engrafted with DP cells exclusively developed CML (Fig. 1e). Giemsa staining of these four fractions showed that both DP and SP cells were blast-like cells, CD27−G$_0$M+ cells were mast-like cells as previously reported[46], and DN cells were differentiated cells (Supplementary Fig. 1d). Moreover, only DP cells could reconstitute the four fractions within CML LSK cells in the BM after secondary transplantation, whereas SP cells could reconstitute three fractions but not the DP fraction (Fig. 1f, g). Imatinib did not affect the characteristics of the four fractions (Fig. 1h–j

and Supplementary Fig. 1e, f). Importantly, imatinib augmented the proportion of DP cells in the CML LSK fraction and the absolute number of DP cells compared with vehicle (Fig. 1c, d). These results demonstrated that CML LSCs were enriched in the DP fraction in the retroviral transduction model and were insensitive to imatinib.

**Upregulation of NF-κB signaling pathways in imatinib-resistant CML LSCs.** To clarify the molecular basis of the LSC

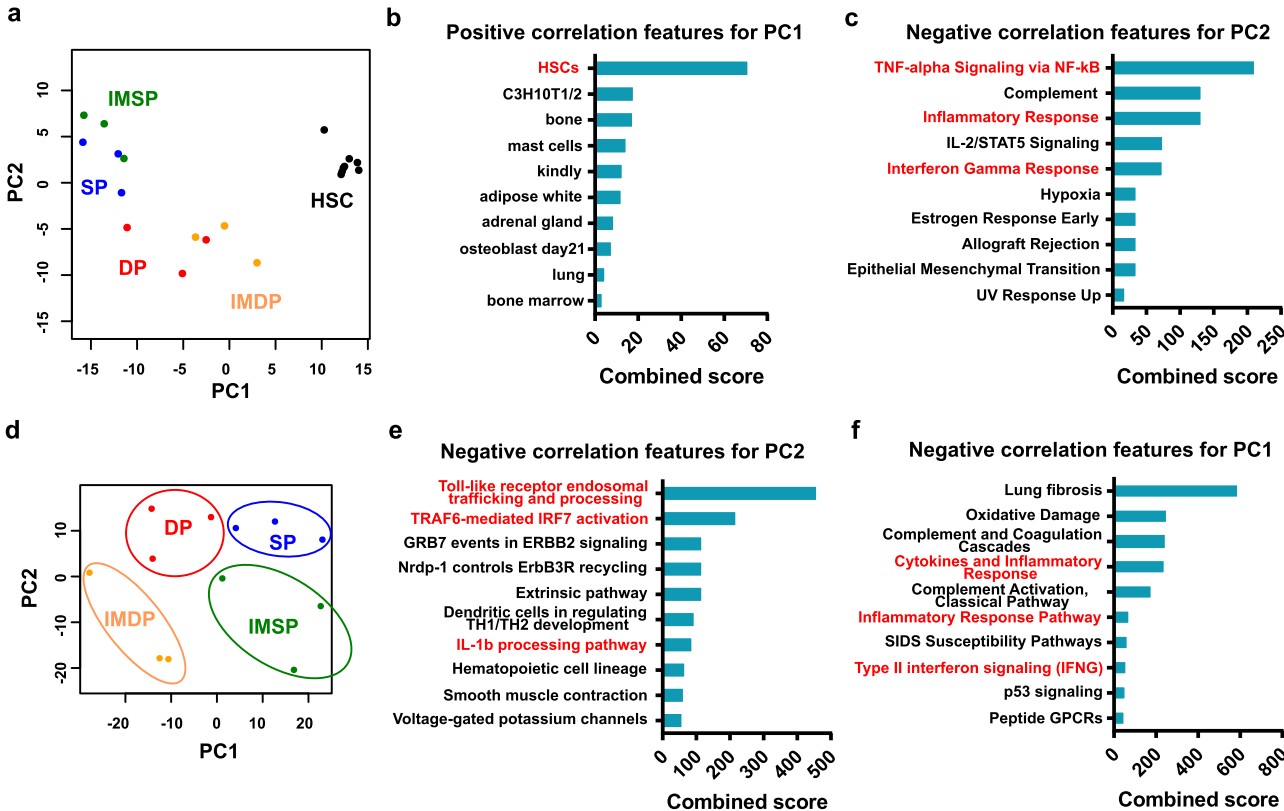

**Fig. 2 Inflammatory signal pathways are enriched in imatinib-insensitive CML-LSCs. a** Principal component analysis (PCA) of the gene expression profiles of the indicated CML populations (SP, DP, IMSP, and IMDP cells: n = 3 each) and normal HSCs (GSE138884) (n = 6). **b, c** Pathway analyses by the Enrichr web tool. Positive correlation features of PC1 (**b**) and negative correlation features of PC2 (**c**) in **a**. **d** PCA of the gene expression profiles of the four different CML populations. **e, f** Pathway analyses by the Enrichr web tool. Negative correlation features of PC2 (**e**) and PC1 (**f**) in **d**.

potential of DP cells, we performed whole-transcriptome (RNA-seq) analysis of SP and DP cells from vehicle- and imatinib-treated CML mice. Principal-component analysis (PCA) and subsequent Enrichr analysis[47] along with a comparison to normal HSCs[44] revealed that DP cells from vehicle-treated mice and DP cells from imatinib-treated mice (IMDP cells) had more HSC features than SP cells from vehicle-treated mice or SP cells from imatinib-treated mice (IMSP cells) (Fig. 2a, b). Representative features for DP and IMDP cells comprised the NF-κB signaling pathway, inflammatory response, IL-2/STAT5 signaling, interferon gamma response, and hypoxia (Fig. 2c). These features are equivalent to CML LSC features, confirming that DP cells represent CML LSCs. Next, we assessed the impact of imatinib on gene expression changes in CML LSCs. PCA clearly separated the four CML populations (Fig. 2d). The subsequent Enrichr analysis showed that negatively correlated features of PC2, which are equivalent to imatinib-resistant features, comprised TLR, TRAF6-mediated, and IL-1β processing pathways, whereas the negatively correlated features of PC1, which are equivalent to CML LSC features, comprised inflammatory response and type II interferon signaling pathways (Fig. 2e, f). The RNA-seq data also revealed that DP cells expressed significantly higher *Tnf* mRNA than normal HSCs and that imatinib treatment attenuated it to near normal HSC levels, indicating that TNFα signaling is not the cause of NF-κB activation in IMDP cells (Supplementary Fig. 2a). In addition, mRNA expression of receptors associated with the non-canonical NF-κB pathway was nearly identical between IMDP cells and HSCs, indicating that the non-canonical NF-κB pathway was also not involved (Supplementary Fig. 2b). These data indicate that signaling pathways related to NF-κB activation via inflammatory signals are crucial for the maintenance of CML LSCs in the presence of imatinib.

**Combination of IRAK1/4 inhibitor and imatinib effectively eliminates CML LSCs.** We next examined whether inflammatory signals in CML LSCs are crucial for the cell survival in the presence of imatinib. TLR1, TLR,2, TLR4, and TLR6 were expressed on CML stem/progenitor cells, and TLR1 and TLR6 were highly expressed on IMDP cells. IL-1R1 was highly expressed on IMSP and IMDP cells (Supplementary Fig. 2c). These data suggested that the inflammatory signals on CML LSCs in the presence of imatinib were transmitted through TLRs/IL1Rs. TLRs except for TLR3 and IL1Rs transduced their signals through Myd88, IRAK1/4, and TRAF6 to NF-κB to activate an inflammatory signaling cascade, and IRAK1/4 inhibitor could block the signals (Fig. 3a). To block TLR/IL1R-NF-κB inflammatory signaling pathways in CML LSCs in the presence of imatinib, we assessed the efficacy of IRAK1/4 inhibitor. We treated CML mice with imatinib, IRAK1/4 inhibitor or the combination of these. The combination greatly prolonged the survival of CML mice and reduced the splenomegaly of CML mice, whereas IRAK1/4 inhibitor alone did not (Fig. 3b, c). In addition, the imatinib-induced increase of the absolute number of DP cells in the BM from CML mice was suppressed by the combination (Fig. 3d, e). Moreover, the combination deactivated NF-κB in DP cells, whereas neither imatinib nor IRAK1/4 inhibitor alone did (Fig. 3f, g). These results demonstrated that NF-κB activation via inflammatory signals was crucial for the survival of CML LSCs in the presence of imatinib, and NF-κB activation via either BCR-ABL1 or inflammatory signals was enough to maintain CML LSCs. Collectively, the combination of IRAK1/4 inhibitor and imatinib is an attractive therapeutic strategy for the elimination of imatinib-insensitive CML LSCs.

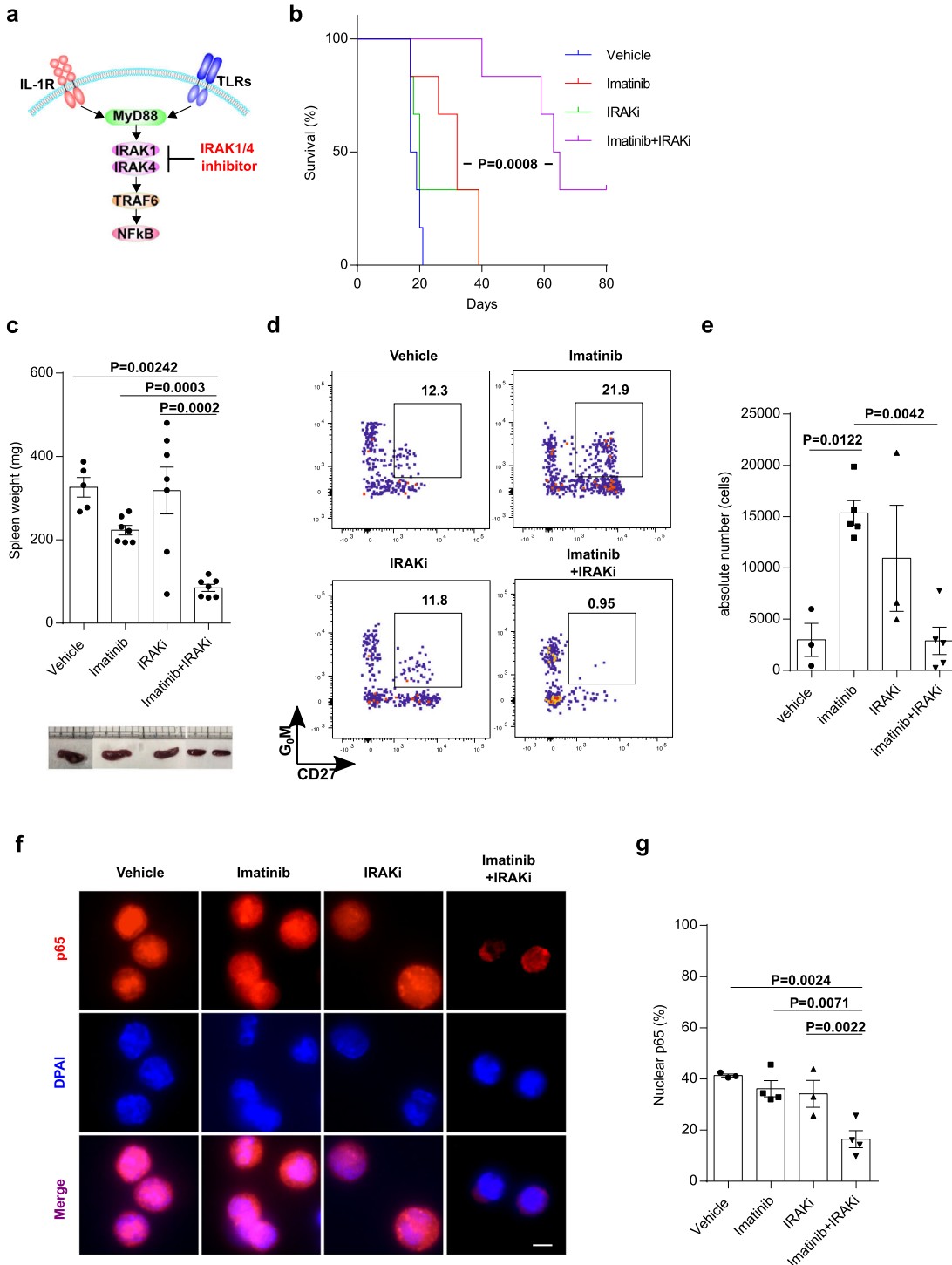

**Fig. 3 IRAK1/4 inhibitor together with imatinib eliminates CML LSCs. a** The IL1R/TLR-IRAK1/4 signaling pathway. **b** Survival curves for mice treated with vehicle, imatinib, IRAK1/4 inhibitor (IRAKi (R930259 Diet): 100 ppm) and the combination ($n = 6$ each). **c** Spleen weights of CML mice treated with vehicle ($n = 5$), imatinib ($n = 7$), IRAKi ($n = 7$), and the combination ($n = 7$). Representative spleens are shown in the bottom. **d** Representative FACS plots of BCR-ABL1$^+$Lin$^-$Sca1$^+$c-Kit$^+$ cells in the BM for all treatment conditions. **e** Absolute number of LSCs (G$_0$M$^+$CD27$^+$ cells in FACS plots in D) in the BM from CML mice treated with vehicle ($n = 3$), imatinib ($n = 5$), IRAKi ($n = 3$), and the combination ($n = 5$). **f** p65 localization in sorted DP cells as CML LSCs for all treatment conditions. **g** Quantification of nuclear p65 in sorted DP cells as CML LSCs from CML mice treated with vehicle ($n = 3$), imatinib ($n = 4$), IRAKi ($n = 3$), and the combination ($n = 4$). Scale bar, 5 μm. All treatments were continued during the observation period. All measurements in **c–g** were performed on days 18–20 after transplants. Data are shown as the mean ± SEM. $P$ values are calculated using log-rank test (**b**) or one-way ANOVA with Tukey's correction for multiple comparisons (**c**, **e**, **g**).

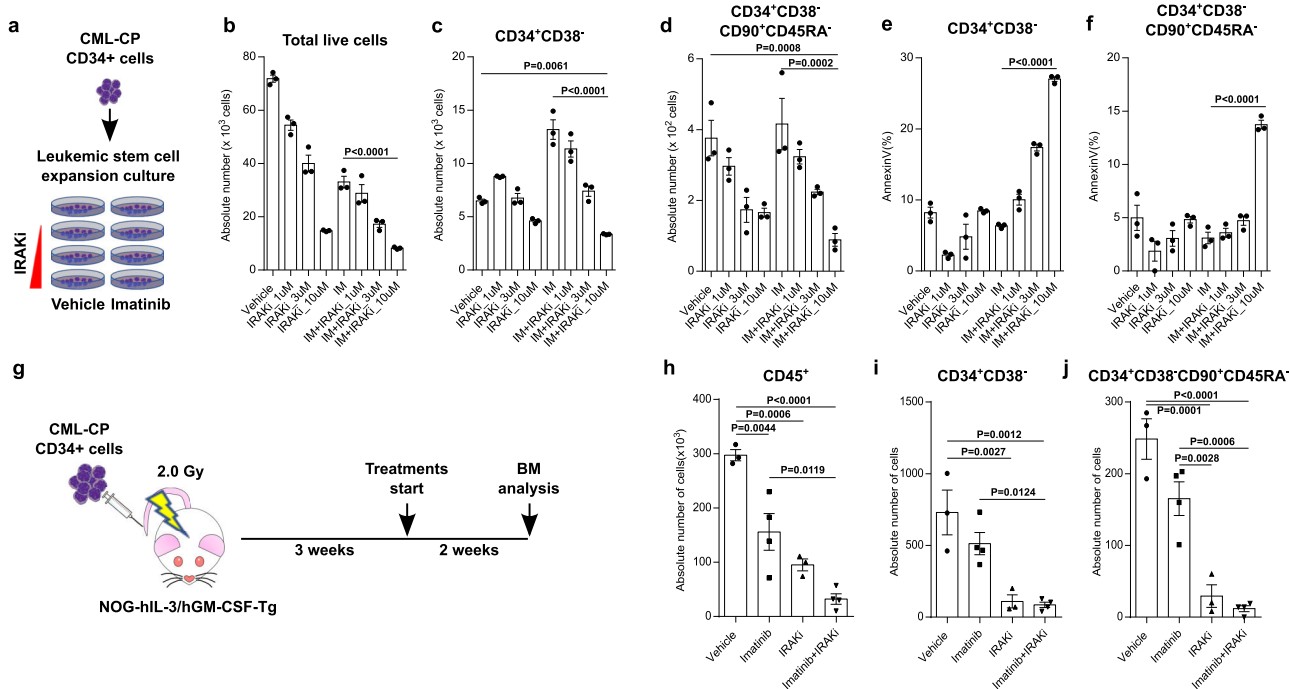

**Fig. 4 IRAK1/4 inhibitor together with imatinib is detrimental to human CML LSCs. a–f** CD34[+] cells (1×10[3]) from the BM of a newly diagnosed CML-CP patient were cultured in liquid culture in the presence of imatinib (IM: 0.5 μM) or IRAK1/4 inhibitor (IRAKi (R568): 1, 3 or 10 μM) alone or in combination for one week (n = 3 each) (**a**). Similar results from two different patients are shown in Fig. S3. Absolute number of total live cells (**b**), CD34[+]CD38[−] cells (**c**) and CD34[+]CD38[−]CD90[+]CD45RA[−] cells (**d**). Percentage of apoptotic cells in CD34[+]CD38[−] cells (**e**) and CD34[+]CD38[−]CD90[+]CD45RA[−] cells (**f**) was measured by annexin staining. **g–j** NOG-hIL-3/hGM-CSF mice were transplanted with CD34[+] cells from the BM of a CML-CP patient (n = 1) and treated with vehicle (n = 3), imatinib (n = 4), IRAK1/4 inhibitor (IRAKi (R930259 Diet): 100 ppm) (n = 3), or the combination of these (n = 4). **g** Experimental design. Absolute number of human CD45[+] cells (**h**), CD34[+]CD38[−] cells (**i**), and CD34[+]CD38[−]CD90[+]CD45RA[−] cells (**j**) in the BM were analyzed at the end of the experiment. Data are shown as the mean ± SEM. P values are calculated using one-way ANOVA with Tukey's correction for multiple comparisons (**b–f**, **h–j**).

## Combination of IRAK1/4 inhibitor and imatinib is effective at reducing human CML LSCs.

We extended the examination of the efficacy of IRAK1/4 inhibitor on CML LSCs to newly diagnosed CML-CP patient samples. CML CD34[+] cells were isolated from the BM mononuclear cells of CML-CP patients (Supplementary Table 1) by immunomagnetic column separation. The CML CD34[+] cells were subjected to leukemic stem/progenitor cell expansion culture in the presence of imatinib (0.5 μM), increasing concentrations of IRAK1/4 inhibitor or the combination of these for 7 days (Fig. 4a). The absolute number of total cells, CD34[+]CD38[−] cells that corresponded to hematopoietic progenitor cells and CD34[+]CD38[−]CD45RA[−]CD90[+] cells that corresponded to HSCs was compared after 7 days culture. A dose-dependent impairment on the proliferation of total cells, CD34[+]CD38[−] cells and CD34[+]CD38[−]CD45RA[−]CD90[+] cells was observed in the presence of IRAK1/4 inhibitor, but the impairment was more effective in combination with imatinib (Fig. 4b–d and Supplementary Fig. 3a–d). Additionally, the increasing concentration of IRAK1/4 inhibitor in the presence of imatinib increased the proapoptotic activity (Fig. 4e, f). Next, we transplanted CML CD34[+] cells into sublethally irradiated immunocompromised NOG mice constitutively producing human interleukin-3 (IL-3) and granulocyte/macrophage-colony stimulating factor (GM-CSF) (NOG hIL-3/GM-CSF Tg[48]) to assess the efficacy of the combination of IRAK1/4 inhibitor and imatinib on the elimination of CML LSCs in a xenograft model with CML-CP patient samples (Fig. 4g). The combination eliminated CD45[+] cells more effectively than imatinib alone (Fig. 4h). IRAK1/4 inhibitor alone and the combination exhibited a potent efficacy on the elimination of CD34[+]CD38[−] cells and

CD34[+]CD38[−]CD90[+]CD45RA[−] cells compared with imatinib alone (Fig. 4i, j). There was a tendency, though not statistically significant, for the combination to be more potent than IRAK1/4 inhibitor alone. We also assessed whether imatinib affects IRAK1 activity and whether IRAK1/4 inhibitors affect BCR-ABL1 activity in K562 cells. Imatinib did not affect levels of either pIRAK1 or IRAK1 (Supplementary Fig. 3e). IRAK1/4 inhibitors did not affect levels of either pCRKL or CRKL as one of readout of BCR-ABL1 activity (Supplementary Fig. 3f). The combination showed slightly stronger inhibition of both pIRAK1 and pCRKL compared to single treatments. These results indicated that IRAK1/4 inhibitors in combination with imatinib has strong efficacy in eliminating human CML-CP LSCs.

## PD-L1 blockade together with imatinib is effective at eliminating CML LSCs.

Our RNA-Seq analysis showed that the expressions of immune checkpoint molecules such as PD-L1, CD80/86, and TIM-3 were relatively higher in IMDP cells than other cell types (Fig. 5a and Supplementary Fig. 4a–c), with PD-L1 mRNA expression highest among them. DP and IMDP cells showed a higher protein level of PD-L1 than SP and IMSP cells, respectively, based on the flow cytometry analysis (Fig. 5b). These data indicated the possibility that PD-L1 is involved in the survival of CML LSCs. PD-L1 expression is regulated by NF-κB via inflammatory signaling such as IL1R, TLR, and IFN signals[49], which were linked to the imatinib-insensitivity of CML LSCs in our RNASeq analysis. The combination of imatinib and IRAK1/4 inhibitor, which deactivated NF-κB, attenuated PD-L1 expression on mouse and human CML LSCs compared with imatinib or

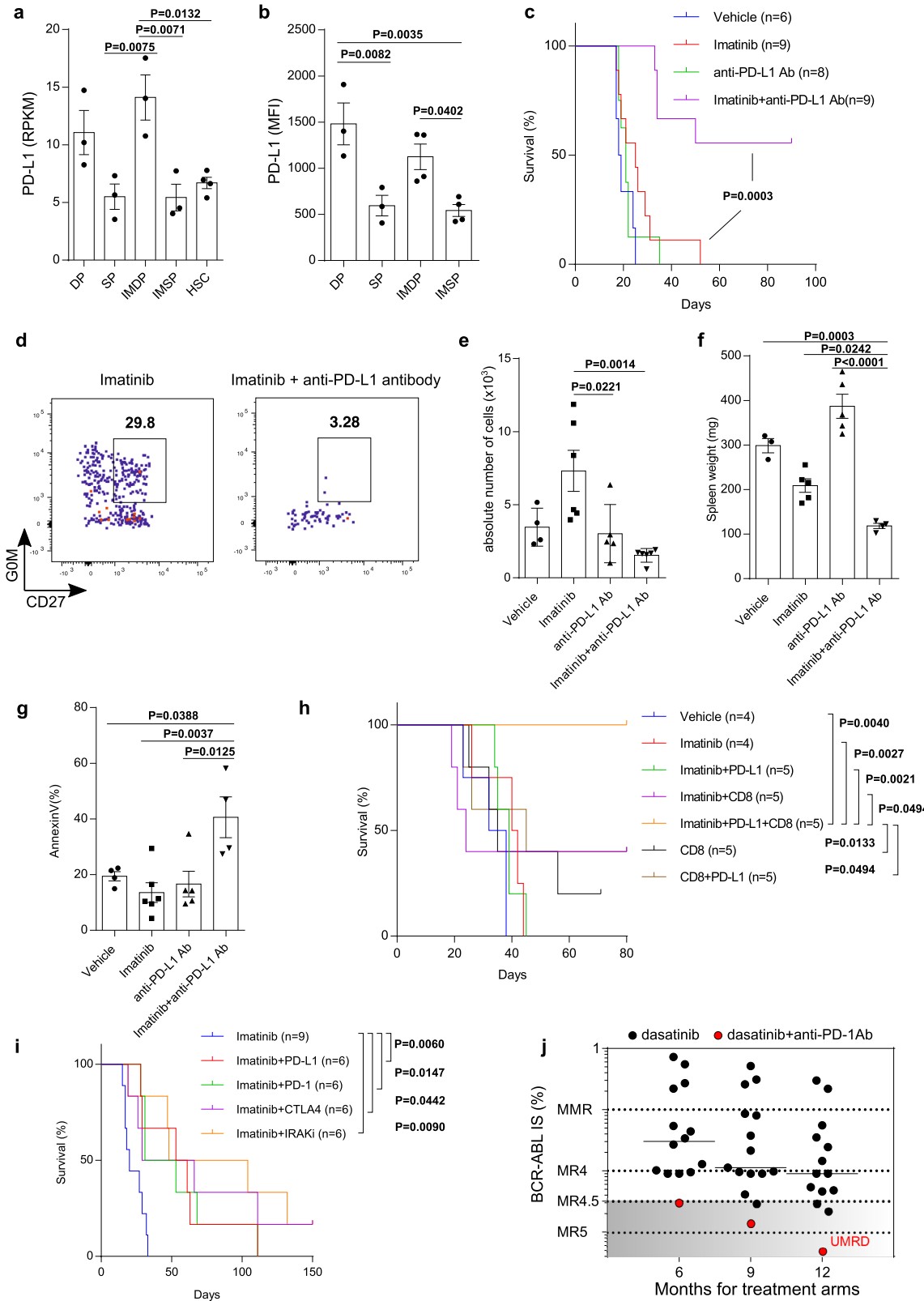

IRAK1/4 inhibitor alone (Supplementary Fig. 4d–f). These observations encouraged us to set up a combination therapy of imatinib and anti-PD-L1 antibody. We treated CML mice with imatinib, anti-PD-L1 antibody, or the combination of these. The combination greatly prolonged the survival of CML mice compared with imatinib or anti-PD-L1 antibody alone (Fig. 5c). The combination also reduced the absolute number of DP cells and splenomegaly compared with imatinib alone (Fig. 5d–f). In particular, the combination therapy-induced apoptosis in CML LSCs (Fig. 5g). Further, PD-L1 expression on CML LSCs from CML mice with the combination treatment was significantly attenuated compared with imatinib alone (Supplementary Fig. 4g). The combination efficacy was abolished when Rag2KO mice[50], which lack T cell immunity, were used as recipients in the mouse

**Fig. 5 The blockade of PD-L1 together with imatinib eliminates CML LSCs. a** RPKM values for PD-L1 among the indicated populations ($n = 3$ (DP, SP, IMDP, and IMSP) and 4 (HSC)). **b** MFI of PD-L1 among the indicated populations ($n = 3$ (DP and SP) and 4 (IMDP and IMSP)). **c** Survival curves for mice treated with vehicle ($n = 6$), imatinib ($n = 9$), anti-PD-L1 antibody ($n = 8$) and the combination ($n = 9$). **d** Representative FACS plots of BCR-ABL1[+]Lin[-] Sca1[+]c-Kit[+] cells in the BM from imatinib and the combination-treated groups. **e** Absolute number of LSCs ($G_0M^+CD27^+$ cells in the FACS plots in **d**) in the BM from all treatment groups ($n = 4$ (Vehicle), 6 (Imatinib), 5 (anti-PD-L1Ab), and 6 (Imatinib and anti-PD-L1 Ab)). **f** Spleen weights of CML mice from all treatment groups ($n = 3$ (Vehicle), 6 (Imatinib), 6 (anti-PD-L1Ab), and 5 (Imatinib and anti-PD-L1 Ab)). **g** The percentage of apoptotic cells in CML LSCs from all treatment groups was measured by annexin staining ($n = 4$ (Vehicle), 6 (Imatinib), 5 (anti-PD-L1Ab), and 4 (Imatinib and anti-PD-L1 Ab)). **h** Survival curves for Ly5.1 Rag2 knockout CML mice treated with vehicle, imatinib, anti-PD-L1 antibody (PD-L1), $2 \times 10^6$ MACS-purified CD8[+] T cells from Ly5.1 mice (CD8) and the combination ($n = 4–5$). **i** Survival curves for CML mice treated with imatinib alone ($n = 9$) or with one of anti-PD-L1, anti-PD-1 and anti-CTLA4 ($n = 6$ each). **j** BCR-ABL1 International Scale (IS) from CML-CP patients treated with dasatinib alone ($n = 14$) or dasatinib together with anti-PD-1 antibody (nivolumab) ($n = 1$) at 6, 9, and 12 months after commencement of dasatinib treatment. MMR (major molecular response) and molecular responses MR4, MR4.5, and MR5 represent BCR-ABL1 transcript levels of $\leq 0.1\%$, $\leq 0.01\%$, $\leq 0.0032\%$, and $\leq 0.001\%$(IS), corresponding to a 3-log, 4-log, 4.5-log, and 5-log reduction from a standardized baseline, respectively. UMRD, undetectable minimal residual disease (The copy numbers of *BCR-ABL1* and *ABL* are as follows; 7.16 and 263921 (6 month), 3.05 and 244113 (9 month), and 0.00 and 135146 (12 month)). All treatments were continued during the observation period. All measurements in **d–g** were performed at days 18–20 after transplants. Data are shown as the mean ± SEM. *P* values are calculated using one-way ANOVA with Tukey's correction for multiple comparisons (**a**, **b**, **e–g**) or log-rank test (**c**, **h**, **i**).

CML-like model. However, CD8 T cell transfer together with the combination successfully recovered the efficacy in these mice (Fig. 5h). Intriguingly, none of the treatments without CD8 T cell transfer prolonged the survival of CML mice, indicating that T cell immunity was crucial for the eradication of CML LSCs. To prove that the treatment effect with the anti-PD-L1 antibody was due to immune checkpoint blockade and not to antibody-dependent cell-mediated cytotoxicity (ADCC), we also assessed PD-1 and CTLA4 which do not induce ADCC. Both anti-PD-1 and anti-CTLA4 antibodies exhibited comparable efficacy with anti-PD-L1 antibody in combination with imatinib, indicating that T cell immunity is more responsible for the effective elimination of CML LSCs than ADCC (Fig. 5i). The efficacy of blocking the immune checkpoint molecules was equivalent to that of IRAK1/4 inhibitor. These results indicate that CML LSCs express immune checkpoint molecules to protect themselves from T cell antitumor immunity. Thus, inducing T cell antitumor immunity by the blockade of immune checkpoint molecules on CML LSCs together with TKIs is potentially effective for the eradication of CML LSCs. Since the combination of IRAK1/4 inhibitor and imatinib attenuates PD-L1 expression on CML LSCs, the combination also induces T cell antitumor immunity in addition to its proapoptotic activity.

We experienced a 60-year-old male with CML which developed during sunitinib therapy against relapsed renal cell carcinoma for 10 months. The Sokal score was in a middle-risk category (Sokal score: 1.02). The patient was treated successfully with dasatinib. In addition to dasatinib, the patient received continuous nivolumab treatment against relapsed renal cell carcinoma 4 months after the dasatinib commencement (Supplementary Fig. 4h). The patient achieved a molecular response with a 4.5-log reduction in *BCR-ABL1* transcripts from baseline (MR4.5; BCR-ABL1 transcript level $\leq 0.0032\%$ [International Scale]) at 6 months and an undetectable minimal residual disease (UMDR) at 12 months, whereas none of 14 patients treated with dasatinib alone at Juntendo University Hospital achieved a MR4.5 at 6 months, and two with low risk (Sokal score < 0.8) did at 12 months (Fig. 5j and Supplementary Table 2). Additionally, only less than 2.0% of all patients who received dasatinib achieved a MR4.5 at 6 months in the phase III Dasatinib Versus Imatinib Study in Treatment-Naïve Chronic Myeloid Leukemia Patients (DASISION) trial[51]. These results suggest that the combination of dasatinib and nivolumab in this patient happened to result in an early achievement of the deep molecular response (>4 log reduction) in CML.

**Inhibition of IRAK1/4-NF-kB-PD-L1 pathway effectively decreases CML-inducing activity of CML LSCs.** We finally

assessed whether the reduction of DP cells by inhibition of IRAK1/4-NF-κB-PD-L1 pathway by either IRAK1/4 inhibitors or anti-PD-L1 antibody in combination with imatinib results in attenuating CML reconstitution activity by secondary transplantation. To this end, we transplanted the whole BM cells from primary CML mice treated with vehicle, imatinib, a combination of imatinib and IRAK1/4 inhibitor and a combination of imatinib and anti-PD-L1 antibody into lethally irradiated recipient mice (Fig. 6a). The number of DP cells in both combination groups was again greatly reduced compared to that in vehicle and imatinib groups (Fig. 6b). In consistent with this, survival of CML mice transplanted with whole BM cells from both combination groups significantly prolonged compared to that from vehicle and imatinib groups (Fig. 6c). Therefore, the decrease of DP cells was correlated with a decrease in the CML reconstitution activity, indicating that CML LSCs are highly enriched in DP cells. Collectively, the inhibition of IRAK1/4-NF-kB-PD-L1 pathway effectively decreases CML LSCs

## Discussion

CML LSCs are thought to be quiescent. This is one of the properties that make them insensitive to TKIs[52,53]. However, the full portrait of CML LSCs and the mechanisms for TKI-resistance are still enigmatic mainly due to the lack of appropriate methods to identify quiescent CML LSCs. Therefore, development of appropriate models to examine the features of quiescent CML LSCs are important. In this study, using a $G_0$ marker ($G_0M$)[43], we found that quiescent CML LSCs are highly enriched in $G_0M^+CD27^+CML$ LSK cells (DP cells). A transcriptomic analysis also revealed that previously reported CML LSC features such as inflammatory signaling, IL2/STAT5 signaling, IFNγ response, and hypoxia pathways were enriched in DP cells[17,18,46,54–57], ensuring that DP cells are CML LSCs. The identification of CML LSCs by $G_0M$ and CD27 enabled us to directly characterize quiescent CML LSCs. We demonstrated that the number of DP cells was increased by imatinib treatment in a mouse CML-like model, indicating that DP cells are insensitive to imatinib. Of note, signaling pathways related to inflammatory response such as TLRs and IL1β signaling pathways were linked to the imatinib insensitivity and pharmacological inhibition of IRAK1/4, mediators of inflammatory response signaling pathways, enhanced the efficacy on eliminating imatinib-insensitive CML LSCs. Moreover, we found that DP cells expressed significantly higher levels of PD-L1 compared to normal HSCs. Because IRAK1/4 inhibitors attenuated PD-L1 on DP cells and the blockade of PD-L1 was also effective on elimination of DP cells, PD-L1 expression is partially regulated by innate immune

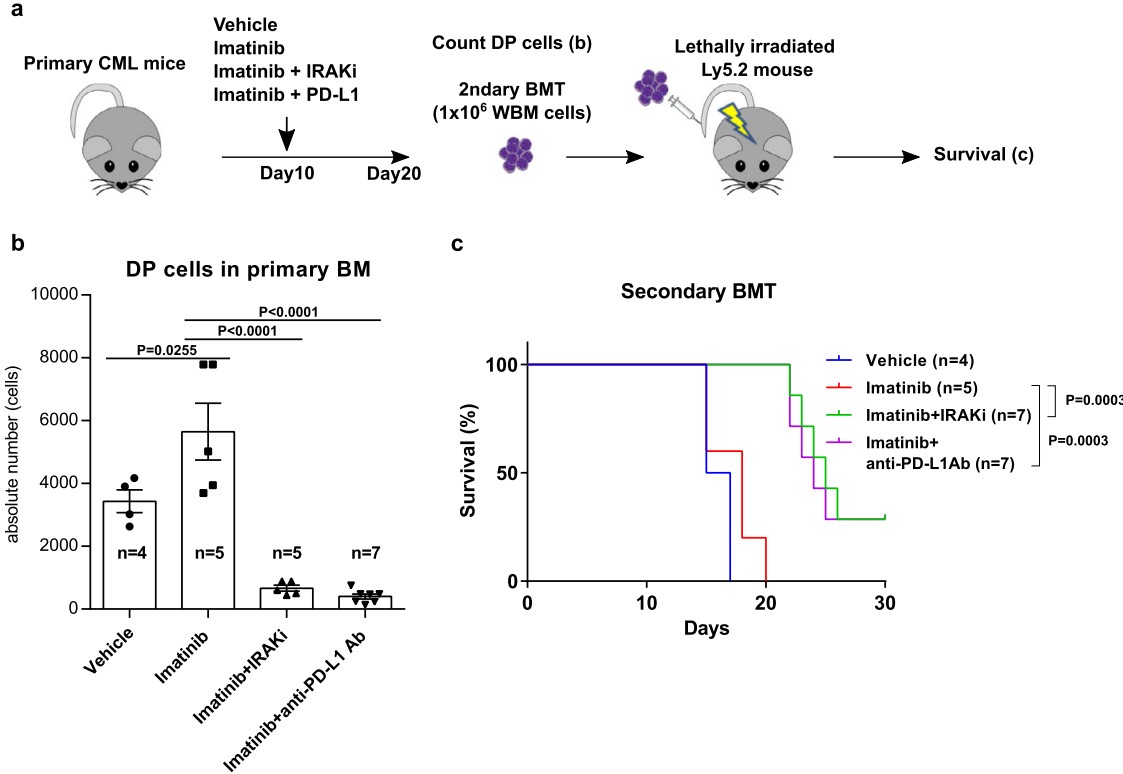

**Fig. 6 A combination of imatinib and IRAK1/4 inhibitors or anti-PD-L1 antibody efficiently reduces the number of CML LSCs. a** Experimental design. **b** Absolute number of DP cells in the BM from primary recipient mice in the indicated treatment groups ($n = 4$ (Vehicle), 5 (Imatinib), 5 (Imatinib + IRAKi), and 7 (Imatinib + PD-L1)). **c** Survival curves for secondary recipient mice transplanted with whole BM cells from primary recipient mice in the indicated treatment groups ($n = 4$ (Vehicle), 5 (Imatinib), 7 (Imatinib + IRAKi) and 7 (Imatinib + PD-L1)). Data are shown as the mean ± SEM. *P* values are calculated using one-way ANOVA with Tukey's correction for multiple comparisons (**b**) or log-rank test (**c**).

stress signals in imatinib-insensitive CML LSCs. Taken together, maintaining PD-L1 expression via inflammatory response signals could be responsible for the insensitivity to imatinib.

Inflammatory response pathways are activated by proteins in cellular stress responses[58]. Melgar et al. showed that activation of inflammatory response pathway via IRAK1 and IRAK4 is essential for adaptive resistance to FLT3 inhibitor in FLT3-mutated AML. Activation of inflammatory response pathways via IRAK1 and IRAK4 are also reported in MDS, AML, T cell acute lymphoblastic leukemia, and lymphoma[35,36,59–62]. In CML, inhibition of IL-1 signaling enhances elimination of TKI–treated LSCs[33,34]. Here we also have shown that IRAK1/4 inhibitors were effective on the elimination of imatinib-treated CML LSCs. Therefore, these data indicate that inflammatory response pathway is also crucial for adaptive response to TKIs in CML LSCs. Imatinib-insensitive CML stem/progenitor cells highly expressed TLR1 and TLR6, which are also overexpressed in MDS BM CD34[+] cells and AML cells[63]. Therefore, the dysregulation of TLRs may be a common adaptive response feature of myeloid leukemias, implicating TLRs as good therapeutic targets. Collectively, inflammatory response pathways are essential for the survival of LSCs in the presence of targeted therapy. However, the precise mechanism of how target therapies activate inflammatory response pathways is not resolved in these studies and our study. Inflammatory response pathways functions through activation of NF-kB and therefore direct inhibition of NF-κB should also work. Indeed, pharmacological blockade of NF-κB activation by an IKK2 inhibitor was reported to be effective on imatinib-resistant CML cells[64]. This indicates that maintaining NF-kB activation is a key mechanism for the survival of LSCs. We showed that neither imatinib nor the IRAK1/4 inhibitor alone deactivated NF-κB,

suggesting that BCR-ABL1 activates NF-κB in the absence of imatinib, whereas IRAK1/4-mediated signals activate NF-κB in the presence of imatinib. Consistently, IL1R/TLR signal pathways were upregulated in CML LSCs in the presence of imatinib. Therefore, upregulation of inflammatory response pathways may work as a surrogate pathway of maintaining NF-kB activation upon targeted therapies.

BM niche is a critical player in both normal hematopoiesis and leukemia[65,66]. CML development induces significant alteration in the BM niche, including markedly altered levels of inflammatory cytokines and chemokines[67], which may enhance LSCs and contribute to TKI resistance[68]. Inflammatory cytokines such as IL-1β from CML cells transform an endosteal BM niche to leukemic one which expresses *Il-1r1*, *Tnf*, and *Tgfb2* significantly higher than normal one and supports CML progression by abnormal production of inflammatory cytokines such as IL-1β[69]. This suggests that inflammatory response pathways are also upregulated in the leukemic niche. Therefore, although we did not examine how IRAK1/4 inhibitors affect the leukemic niche in this study, pharmacological inhibition of IRAK1/4 would also block inflammation of the BM leukemic niche which may enhance leukemic transformation. TLRs should also be involved in this positive feedback of inflammatory response pathways between CML cells and the leukemic niche, but the roles of TLRs in this context is still largely unknown.

CML is considered to be one of the cancers most sensitive to immunological manipulation[70]. One of the escape mechanisms of CML LSCs from immunological attack is the expression of immune checkpoint molecules. We here demonstrate that PD-L1 expression on CML LSCs is higher than that on CML progenitors, consistent with a previous study[41]. PD-L1 expression on human

CML-LSCs (Lin-CD34$^+$CD38$^{-/low}$) is also higher than that on the same population of healthy controls[23,71]. In addition, blocking the PD-1/PD-L1 interaction by PD-L1 or PD-1 antibody together with cytotoxic T cell (CTL) transfer eradicates CML LSCs[41,42], indicating that PD-L1 protects CML LSCs from CTL-mediated elimination. Consistent with this, we identified that the combination of imatinib and anti-PD-L1 antibody effectively eliminated CML LSCs in the mouse CML-like model. In addition, the efficacy of the blockade of the PD-L1/PD-1 interaction was abolished in the absence of T cell immunity. Importantly, both anti-PD-1 and anti-CTLA4 antibodies in combination with imatinib were also effective at prolonging the survival of CML mice, suggesting that immune checkpoint molecules are crucial for the protection of CML LSCs against T cell antitumor immunity. These results indicate that blocking the interactions of immune checkpoint molecules, PD-1/PD-L1 or CTLA4/CD80/CD86 interactions, together with TKIs is effective at eliminating human CML LSCs in the presence of T cell immunity. These results also suggest that both PD-L1 and CD80/CD86 are required for the protection of CML LSCs. Here we reported one clinical case who received dasatinib for newly diagnosed CML-CP and nivolumab for the existing relapsed renal cell carcinoma. This patient exhibited an early MR4.5 achievement compared with dasatinib alone at Juntendo University Hospital and in the DASSISION trial. However, since some studies have reported that approximately 20% of patients with dasatinib exhibited a MR4.5 achievement by 6 months[72,73], this case was not unheard of. While no conclusion can be drawn from a single case, it is tempting to think that combination therapy between TKIs and anti-PD-L1 antibodies would be beneficial for CML patients. In addition, since dasatinib has been reported to have immuno-suppressive effects and the immune properties of this combination therapy may be different from those of imatinib, this case may not fully support our experimental results. Ongoing trials using checkpoint inhibitors in combination with TKIs (imatinib, dasatinib, nilotinib, and bosutinib)[74] will shed light on this discrepancy. However, at present, it would not be practical to use checkpoint inhibitors for patients with CML-CP because TKI alone is able to efficiently extend the survival of the patients without severe side effects but on the contrary checkpoint inhibitors may cause severe side effects. Therefore, safer strategies for the blockade of PD-1/PD-L1 other than PD-1 or PD-L1 antibodies are needed.

Intriguingly, the IRAK1/4 inhibitor in combination with imatinib attenuated PD-L1 expression on mouse and human CML LSCs. Since PD-L1 expression is regulated by NF-κB[75,76], the deactivation of NF-κB by the combination resulted in the attenuation of PD-L1 expression on CML LSCs. Therefore, a part of the efficacy of the IRAK1/4 inhibitor should be mediated by activation of T cell antitumor immunity via the attenuation of PD-L1 expression. Moreover, since IRAK1/4 inhibitors induced apoptosis in human CML LSCs in the absence of T-cell immunity, it is likely that the inhibitors themselves also directly induce apoptosis in CML LSCs. Collectively, IRAK1/4 inhibitors appear to have at least two functions in the elimination of CML LSCs. On the other hand, Suzuki et al. showed that IRAK4 signaling is also essential for eliciting adaptive immune responses[75]. Therefore, IRAK1/4 inhibitor may have some negative effects on T cell immunity. However, considering that the combination of IRAK1/4 inhibitor and imatinib was effective at eliminating CML LSCs in the presence of T cell immunity, the negative effect of IRAK1/4 inhibitor may be minor in our experiments. PD-L1 has been reported to directly enhance cancer cell survival and tumor progression, in addition to its immunosuppressive function[77–79], indicating that the PD-L1 blockade itself is detrimental to CML LSCs. However, because PD-L1 blockade was not effective at

eliminating CML LSCs in the absence of T cell immunity, the proapoptotic function of the IRAK1/4 inhibitor is independent of the attenuation of PD-L1.

In summary, we developed a method for detecting quiescent CML LSCs using G$_0$M in a mouse CML-like model. The identification of quiescent CML LSCs enabled drug screening directly against quiescent CML LSCs. Using this system, we found that an IRAK1/4 inhibitor in combination with imatinib eliminated CML LSCs by dual functions. The one is proapoptotic and the other is the induction of antitumor immunity by attenuating immune check point molecules. Therefore, IRAK1/4 inhibitors are attractive drugs in eliminating CML LSCs with TKIs. However, since IRAK1/4 are central mediators of inflammatory responses, optimization of the drug dose is crucial for clinical use.

## Methods

**Reagents**. Imatinib was purchased from LC Laboratories. IRAK1/4 inhibitor (R568 for in vitro experiments and R930259 Diet (100 ppm) for in vivo experiments) was provided by Rigel Pharmaceuticals, Inc.

**Mice**. C57BL/6J mice were purchased from Japan SLC. G$_0$M mice (*Vav1-Cre*; *Rosa$^{R26R-G0M/ R26R-wt}$* [B6.Cg-Gt(ROSA)26Sor<tm1(mVenus/Cdkn1b<*>)Toki>, BRC No. 109200]) mice were obtained by in-house breeding. Rag2 knockout (B6(Cg)-Rag2tm1.1Cgn/J) mice were purchased from The Jackson Labo Service Corporation. NOG hIL-3/GM-CSF Tg (NOD.Cg-Prkdcscid Il2rgtm1Sug Tg(SRa-IL3, CSF2)7-2/Jic) mice were provided by the Central Institute for Experimental Animals (CIEA; Japan). All animal studies were approved by the Animal Care Committee of the Institute of Medical Science at the University of Tokyo (approval number: PA13-19 and PA16-31) and were conducted in accordance with the Regulation on Animal Experimentation at the University of Tokyo based on International Guiding Principles for Biomedical Research Involving Animals.

**Patient samples**. Proper informed consent was obtained from all participants. All experiments were performed according to the Declaration of Helsinki and were approved by the Ethics Committee of Juntendo University School of Medicine (IRB#2019113) and the Research Ethics Review Committee of the Institute of Medical Science of the University of Tokyo (2019-8-0702). Heparin-anticoagulated BM cells were obtained from 5 newly diagnosed CML-CP patients (Table S1). Informed consent was obtained in accordance with the Declaration of Helsinki, and all procedures were approved by the Research Ethics Board at Juntendo University. Mononuclear cells were isolated using Lymphoprep (STEMCELL Technologies) density gradient separation, and CD34$^+$ cells (>85%) were enriched immunomagnetically using an CD34 MicroBead Kit (Milteny Biotec). Purity was verified by re-staining isolated cells with an allophycocyanin-cyanine7-labeled (APC-Cy7) anti-CD34 antibody (clone 561, BioLegend), and the cells were analyzed using a FACSAria (BD) machine.

**Preparation of retrovirus**. The cDNA encoding human BCR-ABL1 (a gift from H.Honda, Tokyo Women's Medical University) was cloned into the EcoRI site of the MSCV-IRES-GFP vector. Retroviral packaging cells (Plat-E) were transiently transfected with the MSCV-BCR-ABL1-IRES-GFP plasmid using polyethylenimine (Sigma-Aldrich) and used for transplantation into mice as described later.

**Generation of a mouse CML-like model**. G$_0$M mice were treated with 5FU and four days later culled to isolate whole BM cells. The cells were cultured in IMDM medium (Thermo Fisher Scientific) supplemented with 20% FBS, 1% penicillin/streptomycin, 50 μM ß-mercaptoethanol, 10 ng/ml mouse interleukin (IL)−6, 50 ng/ml human TPO, and 50 ng/ml mouse stem cell factor (SCF) overnight. On the next day, the cells were infected with the above retrovirus carrying MSCV-BCR-ABL1-IRES-GFP using RetroNectin®□ (TaKaRa Bio Inc.). In all, $1 \times 10^3$ GFP-positive cells were transplanted intravenously into lethally irradiated (9.5 Gy) C57BL/6 mice together with $1 \times 10^5$ BM mononuclear cells from non-irradiated C57BL/6 mice.

**Flow cytometric analysis and antibodies**. BM hematopoietic cells were isolated from the femurs and tibias by flushing and the depletion of red blood cells by ACK Lysing Buffer (Thermo Fisher Scientific). The cells were stained with the appropriate dilution of fluorochrome- and/or biotin-conjugated antibodies and 4′,6-diamidino-2-phenylindole (DAPI) or propidium iodide (PI) for dead cell exclusion and analysis using a FACSVerse flow cytometer and FASCAria (BD Biosciences). The following antibodies (all BioLegend) were used: Lineage Cocktail [CD5(53-7.3)-biotin, TER-119(TER-119)-biotin, CD11b(M1/70)-biotin, Gr-1(RB6-8C5)-biotin and B220(RA3-6B2)-biotin, 1:1000 each] with streptavidin BV605 (1:500), c-Kit(2B8)-PE-Cy7(1:500), Sca-1(D7)-APC(1:500) and CD27(LG.3A10)-BV421 (1:500) for mouse CML LSC analysis; CD34(561)-APC-Cy7(1:500),

CD38(HB-7)-PE-Cy7(1:500), CD90(5E10)-biotin(1:500) with streptavidin BV605(1:500), CD45RA(HI100)-FITC(1:500), CD45(2D1)-FITC(1:250) or -BV421(1:250), Lineage Cocktail [CD3(OKT3), CD14(M5E2), CD16(3G8), CD19(HIB19), CD20(2H7), CD56(HCD56), 1:100]-BV510 for human CML LSC analysis; and CD274(10F.9G2 and 29E.2A3)-PE(1:500) for the detection of mouse and human PD-L1, respectively. The detection of annexin V-positive cells for apoptosis analysis was performed using either Annexin V-PE or APC.

**In vivo treatment of mice with CML-like disease**. All treatments were commenced 10 days after the BM transplantation. Imatinib (40 mg/kg) or vehicle was administered once daily by oral gavage. The IRAK1/4 inhibitor was administered by feed administration (R930259 Diet (100 ppm)). Anti-PD-L1 antibody (10 F.9G2, BioXcell; 200 μg), anti-PD-1 antibody (29 F.1A12, BioXcel; 200 μg), anti-CTLA-4 antibody (9D9, BioXcel; 200 μg), or their isotype control antibodies were administered by intraperitoneal injection once a week. For the CD8 transfer, $2 \times 10^6$ CD8 T cells from the spleen were intravenously injected into CML mice.

**Secondary transplantation**. In all, $1 \times 10^6$ whole BM cells of primary recipient mice from each treatment group were transplanted intravenously into lethally irradiated (9.5 Gy) C57BL/6 mice.

**In vitro drug treatment for patient samples**. In all, $1 \times 10^2$ or $1 \times 10^3$ CML-CP CD34+ cells were cultured using the StemSpan™ Leukemic Cell Culture Kit (STEMCELL Technologies) in a well of a 96-well flat-bottom plate in the presence of DMSO, imatinib (0.5 μM), IRAK1/4 inhibitor (R568; 1, 3, and 10 μM), or the combination of for 7 days. At day 7, all cells were subjected to flow cytometric analysis to count the number of live cells and assess the proportion of CD34+CD38- and CD34+CD38−CD90+CD45RA+ fractions. Measurements of Annexin V positive cells in those fractions were performed at the same time.

**Immunodeficient murine xenograft experiments**. CML CD34+ cells were transplanted ($1 \times 10^5$ or $1 \times 10^6$ cells per mouse) via tail-vein injection into female 8-to-12-week-old sublethally irradiated (2.0 Gy) NOG hIL-3/GM-CSF Tg mice (CIEA). Human cell engraftment was monitored at 4 to 6 weeks after transplantation by FACS for the presence of human CD45+ cells in the blood. Mice that showed evidence of human cell engraftment were randomized and treated with vehicle (water), imatinib, IRAK1/4 inhibitor in feed (R930259 Diet (100 ppm)), or a combination of imatinib and IRAK1/4 inhibitor for 14 or 21 days commencing at 2 weeks after the transplantation. Imatinib (40 mg/kg) or vehicle were administered once daily by oral gavage. Mice were euthanized at the end of the treatment, and marrow contents of the femurs and tibias were obtained. The presence of human CD45, CD34, CD38, CD90 CD45RA, and CD274 on the engrafted cells was assessed by FACS. Engraftment data were obtained from CML CD34+ samples treated for 21 days from two patients.

**Western blotting analysis**. BCR-ABL1-expressing K562 cells (CCL-243, ATCC) were washed with PBS and lysed with pre-heated Laemmli sample buffer (BIO-RAD, USA; #1610737). Total cell lysates were subjected to western blotting analysis. Signals were detected with ECL Western Blotting Substrate (Promega, USA; #W1001), and the immunoreactive bands were visualized by LAS-4000 Luminescent Image Analyzer (FUJIFILM). The following antibodies were used in this study at indicated dilutions: anti-IRAK1 (Cell Signaling Technology, USA; catalog #4501, clone D51G7, 1:500), anti-phospho IRAK1 (pThr209) (Sigma-Aldrich, JAPAN; catalog #SAB4504246, 1:500), anti-CrkL (Cell Signaling Technology, USA; catalog #3182, clone 32H4, 1:500), anti-Phospho-CrkL (Tyr207) (Cell Signaling Technology, USA; catalog #3181, 1:500), and anti-GAPDH (Cell Signaling Technology, USA; catalog #5174, clone D16H11, 1:1000).

**RNA sequencing**. We performed RNA sequencing (RNA-Seq) as described previously[80] with minor modification. In brief, using 100 sorted cells, the first strand of cDNA was synthesized using a PrimeScript RT reagent kit (TaKaRa Bio Inc.) and not-so random primers. Following the synthesis of the first strand, the second strand was synthesized by Klenow Fragment ($3'$, $−5'$, exo-; New England Biolabs Inc.) and complement chains of the not-so random primers. Using purified double-strand cDNA, the library for RNA-Seq was prepared and amplified using a Nextera XT DNA sample Prep Kit (Illumina Inc.). These prepared libraries were sequenced on a Next-Seq system (Illumina Inc.) according to the manufacturer's instruction. In addition, each obtained read was mapped to the reference sequence "GRCm38/mm10" using CLC genomic workbench v11.0.0 (Qiagen), and expression levels were normalized and subjected to statistical analyses based on EdgeR.

**Statistical analyses**. Statistical differences were determined using the unpaired and two-tailed Student's t-test, one-way ANOVA with Tukey's correction for multiple comparisons, and the log-rank (Mantel-Cox) test for the survival analysis using Graphpad Prizm6 software (MDF).

**Reporting summary**. Further information on experimental design is available in the Nature Research Reporting Summary linked to this paper.

## Data availability
The RNA-sequencing data have been deposited in the NCBI gene expression omnibus under the accession code GSE175323. Source data are provided with this paper.

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

## Acknowledgements

We thank H. Honda for BCR-ABL1 cDNA. This work was supported by the following grants: SENSHIN Medical Research Foundation (Y.T.) and Takeda Science Foundation (Y.T.). We acknowledge the IMSUT FACS Core Laboratory and the IMSUT Animal Research Center.

## Author contributions

Conceptualization and methodology, Y.T. and T.K.; investigation, R.T., K.M., T.Fukushima, K.A., S.A., M.T., T.U., H.T., and Y.T.; writing – original draft, Y.T. and T.K.; writing – review and editing, Y.T., S.G., and T.K.; funding acquisition, Y.T. and T.K.; resources, S.T., N.W., S.M., M.I., M.A., M.N., T.T., N.K., R.I., C.L., and E.S.M.; Supervision, S.G., T.Fukuyama, and T.K.

## Competing interests

The authors declare no competing interests.
