## [Peer Review File · Nature Communications]

Eliminating chronic myeloid leukemia stem cells by IRAK1/4 inhibitorsREVIEWER COMMENTS

Reviewer #1 (Remarks to the Author); expert on immunotherapy and leukaemia:

In this study, Tanaka et al. first provide evidence for a novel G0 marker that helps identifying quiescent LSCs for CML in a mouse model. They then used this marker to characterize LSCs by transcriptome analyses, as well as in xenotransplantation models. As they identified NFkB activity and inflammatory signalling as central nodes in these cells, they used an IRAK1/4 inhibitor for preclinical studies in mice and showed improved treatment efficacy when combined with imatinib. As LSCs showed an induced expression of PD-L1, specifically after imatinib treatment, they explored the role of checkpoint inhibitors in the CML mouse models.

The study is highly interesting with novel and convincing data presented in a very clear and concise manner. The manuscript is very well written, except of the discussion, which is in large parts a repetition of data description, even referring again to the figures. Rewriting of the discussion and avoiding a repetition of results description would be of great benefit for the manuscript.

My main criticism is concerning the last part of the study involving immune checkpoint blockade. As imatinib induces PD-L1 expression in LSCs, treatment with anti-PD-L1 likely results in ADCC, mediating a depletion of LSCs via the immune system. This is also supported by the T-cell dependency that is shown. As the authors also performed PD-1 treatment in their model, showing the results of this study in the main figure would be of higher interest, since here the assumption that the treatment effect is due to immune checkpoint blockade instead of ADCC is given, which is not the case for anti-PD-L1.

In addition, lack of PD-L1 detection on LSCs of mice treated with anti-PD-L1 by flow cytometry might be rather explained by blocking of antibody binding upon anti-PD-L1 treatment and not by a downregulation of the protein itself. Putting the PD-1 data in the focus of the study would be also more in line with the treatment strategy of patients, who also receive anti-PD-1 treatment.

What is the difference between the data shown in Figure 5C and Suppl Figure 4H? PD-L1 treatment seems a lot more effective in the data set shown in the main figure with about 50% of mice still alive after almost 100 days. The authors need to comment on the discrepancy here.

Minor comments:

Fig. 2A and D: titles of axes are not readable.

Mention Fig. 3G in the text in line 204.

Fig. 3F and G: which exact cells are analysed here? Sorted DP cells?

The authors should explain the reasoning for discriminating a subset of CD90+CD45RA- cells in Figure 4, as this might not be clear to the non-experienced reader.

The authors need to explain MMR and MR values shown in Fig. 5J.

Reviewer #2 (Remarks to the Author); expert on CML and LSC:

In this manuscript by Tanaka et al. the authors present an exciting new method of identifying quiescent CML LSC by using a G0 marker and show that LSC are G0M and CD27 double positive. The authors show that the imatinib-insensitive double positive population of LSC exhibits NF-KB activation. Inhibition of NF-KB by IRAK1/4 inhibitors in combination with imatinib led to eradication of LSC in murine models and in in vitro experiments using human CML cells. In addition, PD-L1 expression was reduced by IRAK1/4 inhibitors, leading to the discovery that blockade of PD-L1 and imatinib eradicated LSC, when T cell immunity was functional, which is a fairly novel concept in CML. The authors use several in vitro and in vivo models to investigate the associated signaling pathways and molecules. The identified targets are elegantly targeted alone and in combination with imatinib. Overall, the manuscript is well written. The following questions remain:

1. Figure 1A: It is not clear when imatinib treatment was initiated. Was imatinib given from day 1 post

transplantation or was it started, once the disease was established (which seems the case). The authors should clearly define this in the schematic. Treatment should have been started, when the disease is established.

2. Can the author show tumor burden (WBC/ μ l) or CD11b+ myeloid cells during the course of disease for figure 1E? It would be informative to see if the SP population does induce the disease at all and if the disease eventually dies out. Also, surely, the mice in this figure are not dying of CML any more after day 30? What are the (histo)-pathological features of the disease in the secondary recipients and the primary recipients (considering that the number of transplanted leukemia-initiating cells is quite low)?

3. Figure 3D: At what point during the course of the disease was the measurement done?

4. In order to comment on the LSC-eradicating ability of IRAKi a secondary transplant would be necessary, i.e. from the experiment in figure 3B (and possibly 5C).

5. Does imatinib treatment affect levels of IRAK1/4? Does IRAK1/4 inhibition compromise BCR-ABL1 activity, i.e. does it affect pCRKL as one readout?

6. Testing T-cell numbers/activity after treatment with IRAKi, also in combination with imatinib would be informative.

7. If the NF-KB pathway is activated in LSC, which ligand/receptor is likely involved? TNF α , TLR? Increased expression of TLRs/IL1Rs in Figure S2 is not really sufficient to prove this question.

8. The concept of targeting IRAK1/4 in hematological malignancies is not entirely novel, e.g.: PMID 29743719 and 32395555. This must be discussed.

9. It is important to consider that any in vivo treatment will also target the bone marrow microenvironment which is likely contributing to the inflammatory state. This should be carefully discussed.

10. Figure S1C: While this reviewer agrees that the cells in this figure (SP and DP) look like blasts, there are clear differences in the size. Also, CML stem cells are not usually described as blasts (and neither are CML cells in chronic phase). Why are the cells in the G0+/CD27- more differentiated than the cells in DN? That seems hard to understand. There is also a discrepancy between text and figure legend about the staining. Surely, these are Giemsa stainings (and not H&E)?

Minor:

At the end of the sentence in the legends the authors frequently write 'combination of', but it should just say 'combination'.

Reviewer #3 (Remarks to the Author); expert on clinical CML:

The authors present interesting data on the potential of IRAK1/4 inhibitors to eradicate early CML progenitor cells. In addition to the cell line and patient cell data, and the animal studies presented, they add the anecdotal evidence from a patient who received dasatinib for CML in addition to nivolumab for a concomitant diagnosis of renal cell cancer. They report an early rapid deep molecular response that the authors suggest may be due to the combination therapy. Both the preclinical and the clinical data are of interest, but warrant some comments:

1. A deep molecular response can hardly be interpreted as eradication of the leukemic stem cells, the presumed benefit of the combination being used. Proof of this is that at least half of patients with sustained deep molecular responses must resume therapy after treatment discontinuation because of loss of major molecular response and yet others have detectable disease albeit at lower levels. Thus, the case being presented is not supportive of the main mechanism being proposed for this combination.

2. Undetectable disease cannot be evaluated unless the control gene is known and the number of copies of it are available for the sample in question. The authors should provide both.

3. The response of the patient was indeed rapid, but not unheard of. Some studies (e.g., Maiti et al. Cancer 2020; Naqvi et al., Cancer 2020) have reported MR4.5 in approximately 20% of patients by 6 months, making this one instance not at all outside of the confidence interval of possibilities.

4. All the preclinical data presented relates to imatinib, whereas the patient in question is treated with dasatinib. One cannot assume the effects can be fully extrapolated. Dasatinib for example has been

reported by some to have immunosuppressive effects that may make the immune properties of this combination different from those of imatinib.

5. Trials using the combination of checkpoint inhibitors and TKIs have been conducted (e.g., NCT02011945). This should at least be acknowledged even if data is not available at this time.

Point-by-point response to the reviewers' comments

Reviewer #1

In this study, Tanaka et al. first provide evidence for a novel G0 marker that helps identifying quiescent LSCs for CML in a mouse model. They then used this marker to characterize LSCs by transcriptome analyses, as well as in xenotransplantation models. As they identified NFkB activity and inflammatory signalling as central nodes in these cells, they used an IRAK1/4 inhibitor for preclinical studies in mice and showed improved treatment efficacy when combined with imatinib. As LSCs showed an induced expression of PD-L1, specifically after imatinib treatment, they explored the role of checkpoint inhibitors in the CML mouse models.

The study is highly interesting with novel and convincing data presented in a very clear and concise manner. The manuscript is very well written, except of the discussion, which is in large parts a repetition of data description, even referring again to the figures. Rewriting of the discussion and avoiding a repetition of results description would be of great benefit for the manuscript.

Thank you for you're the positive comments for our manuscript and a valuable suggestion in improving it. According to your comment, we have rewritten the discussion.

My main criticism is concerning the last part of the study involving immune checkpoint blockade.

As imatinib induces PD-L1 expression in LSCs, treatment with anti-PD-L1 likely results in ADCC, mediating a depletion of LSCs via the immune system. This is also supported by the T-cell dependency that is shown. As the authors also performed PD-1 treatment in their model, showing the results of this study in the main figure would be of higher interest, since here the assumption that the treatment effect is due to immune checkpoint blockade instead of ADCC is given, which is not the case for anti-PD-L1.

In addition, lack of PD-L1 detection on LSCs of mice treated with anti-PD-L1 by flow cytometry might be rather explained by blocking of antibody binding upon anti-PD-L1 treatment and not by a downregulation of the protein itself. Putting the PD-1 data in the focus of the study would be also more in line with the treatment strategy of patients, who also receive anti-PD-1 treatment.

What is the difference between the data shown in Figure 5C and Suppl Figure 4H? PD-L1 treatment seems a lot more effective in the data set shown in the main figure with about 50% of mice still alive after almost 100 days. The authors need to comment on the discrepancy here.

Thank you for your valuable comments and suggestions. According to the reviewer's comment, we have moved the PD-1 data to the main figure (a new Figure 5I) although we had only survival data of PD-1 treatment. We have also added descriptions in the result section according to your recommendation (lines 275-280).

The difference between the two experiments is probably due to the difference in the intensities of the CML-like disease which vary depending on the experiments when the retrovirus-induced mouse CML-like model is used, because we have always used the same protocol. Nevertheless, the combination of imatinib and anti-PD-L1 antibody significantly prolonged survival of CML mice compared to imatinib alone in these two independent experiments.

Minor comments:

Fig. 2A and D: titles of axes are not readable.

We have corrected them.

Mention Fig. 3G in the text in line 204.

We have added Fig.3G in the new text in line 212.

Fig. 3F and G: which exact cells are analysed here? Sorted DP cells?

The cells analyzed were sorted DP cells. We have added the details of the cells in the figure legend (lines 641 and 642).

The authors should explain the reasoning for discriminating a subset of CD90+CD45RA- cells in Figure 4, as this might not be clear to the non-experienced reader.

We have added the explanation of CD34+CD38- cells and CD34+CD38-CD90+CD45RA- cells in the new text in lines 226 and 227.

The authors need to explain MMR and MR values shown in Fig. 5J.

We have added the explanation of MMR, MR4, MR4.5 and MR5 in the figure legend in lines 686-689.

Reviewer #2 (Remarks to the Author); expert on CML and LSC:

In this manuscript by Tanaka et al. the authors present an exciting new method of identifying quiescent CML LSC by using a G0 marker and show that LSC are G0M and CD27 double positive. The authors show that the imatinib-insensitive double positive population of LSC exhibits NF-KB activation. Inhibition of NF-KB by IRAK1/4 inhibitors in combination with imatinib led to eradication of LSC in murine models and in in vitro experiments using human CML cells. In addition, PD-L1 expression was reduced by IRAK1/4 inhibitors, leading to the discovery that blockade of PD-L1 and imatinib eradicated LSC, when T cell immunity was functional, which is a fairly novel concept in CML. The authors use several in vitro and in vivo models to investigate the associated signaling pathways and molecules. The identified targets are elegantly targeted alone and in combination with imatinib.

Overall, the manuscript is well written. The following questions remain:

1. Figure 1A: It is not clear when imatinib treatment was initiated. Was imatinib given from day 1 post transplantation or was it started, once the disease was established (which seems the case). The authors should clearly define this in the schematic. Treatment should have been started, when the disease is established.

Thank you for your comments. We started imatinib treatment 10 days after transplantation when CML was established. We have amended the experimental design in Figure 1.

2. Can the author show tumor burden (WBC/ μ l) or CD11b⁺ myeloid cells during the course of disease for figure 1E? It would be informative to see if the SP population does induce the disease at all and if the disease eventually dies out. Also, surely, the mice in this figure are not dying of CML any more after day 30?

Thank you for your important comments. According to the comments, we have now added kinetics data of CD11b⁺ myeloid cells in Figure S1 and its description in the new text in lines 154-156. The SP cells gave rise to CD11b⁺ CML cells, but they finally died out. All mice except one transplanted with SP cells survived during the observation period (60 days).

What are the (histo)-pathological features of the disease in the secondary recipients and the primary recipients (considering that the number of transplanted leukemia-initiating cells is quite low)?

The secondary CML mice showed the similar features as the primary ones, such as increasing chimerism of CML CD11b⁺ cells as shown in a new Figure S1, splenomegaly (data not shown) and the similar CML KSL profiles as shown in Figure 1F. For secondary transplantation, we have transplanted 3000 DP cells into the secondary recipients, whereas only 1000 5-FU-treated BM cells were used in the primary transplantation. Since survival of secondary CML mice tended to be a little longer than that of primary CML mice, leukemic-initiating cells in DP cells from primary recipients may be quite lower compared to 5-FU-treated BM cells (Figure 1).

3. Figure 3D: At what point during the course of the disease was the measurement done?

The experiment on CML LSCs was performed at day18-20 after treatment, because most of CML mice from the vehicle group became moribund at that time point. We have added this information in the figure legend in line 644.

4. In order to comment on the LSC-eradicating ability of IRAKi a secondary transplant would be necessary, i.e. from the experiment in figure 3B (and possibly 5C).

Thank you very much for raising this important point. We performed secondary transplantation assay using whole bone marrow mononuclear cells from primary recipient mice after 10 days treatment with vehicle, imatinib, imatinib plus IRAK inhibitor and imatinib plus anti-PD-L1 antibody. Survival of secondary recipient mice transplanted with whole bone marrow from imatinib+IRAKi and imatinib+anti-PD-L1 significantly prolonged compared to that from vehicle and imatinib. Therefore, these combinations efficiently reduced the number of CML LSCs, but not eradicated CML LSCs by 10-day treatment. The remaining DP cells even after the combination treatment could reconstitute CML in this setting. Considering that some primary CML mice treated with the combinations have never relapsed CML, probably the anti-tumor T-cell immunity have been established during the treatment. However, once DP cells from these mice were transplanted into new recipients, they reconstituted CML again because the anti-tumor T-cell immunity are not ready in these mice. We have now added new sections in lines 305-318 and figures on secondary transplantation as Figure 6. In this regard, we have also replaced “eradication” with “elimination” to weaken the expression.

5. Does imatinib treatment affect levels of IRAK1/4? Does IRAK1/4 inhibition compromise BCR-ABL1 activity, i.e. does it affect pCRKL as one readout?

Thank you for the critical comment. In response to the comment, we examined whether imatinib affects levels of IRAK1 and whether IRAK1/4 inhibitors affect levels of pCRKL in BCR-ABL1-expressing K562 cells. Briefly, K562 cells were treated with increasing doses of imatinib, IRAK1/4 inhibitors or those

combinations for 2 hours. Then, we measured pIRAK1, IRAK1, pCRKL, or CRKL by immunoblotting. Increasing doses of imatinib (maximum; 5 μ M) did not affect levels of either pIRAK1 or IRAK1. Increasing doses of IRAK1/4 inhibitors did not affect levels of either pCRKL or CRKL. We have added the descriptions in main text in lines 243-248 and the data to Figure S3.

6. Testing T-cell numbers/activity after treatment with IRAKi, also in combination with imatinib would be informative.

Thank you for raising the important point. Unfortunately, long-lived mice treated with IRAKi and IM+IRAKi has been already culled and we have not examined T-cell numbers/activity and we cannot present T cell data. Suzuki et al showed that IRAK-4 signaling is essential for eliciting adaptive immune responses (Suzuki, 2006). Therefore, IRAKi may have some negative effects on T cell immunity. Considering that the combination was effective in B6 mice setting, the negative effect of IRAKi may be minor in our experiments. We have mentioned this in discussion in lines 426-430. We will assess this point in the future study.

Suzuki, N., 2006. A Critical Role for the Innate Immune Signaling Molecule IRAK-4 in T Cell Activation. *Science* 311, 1927–1932.

<https://doi.org/10.1126/science.1124256>

7. If the NF-KB pathway is activated in LSC, which ligand/receptor is likely involved? TNFalpha, TLR? Increased expression of TLRs/IL1Rs in Figure S2 is not really sufficient to prove this question.

Thanks for an important question. It was reported that TNFalpha-NF-kB positive feedback loop is important for CML progression and maintenance of CML LSCs. Our RNASeq data also showed that Tnf mRNA expression in DP cells was much higher than that in normal HSCs and SP cells, indicating that TNFalpha-NF-kB positive feedback loop is activated on DP cells. We also found that imatinib treatment greatly attenuated Tnf (gene symbol) mRNA expression in DP cells close to normal HSC level. This is consistent with the previous study

showing that imatinib inhibits the production of the TNF α (Wolf et al., 2005). Therefore, TNF α -NF- κ B positive feedback loop would not be active in DP cells in the presence of imatinib. We also found that mRNA expression of IL-1R1 and TLR6 was significantly higher in imatinib-insensitive DP cells compared to that in normal HSCs. The expression of other relevant receptors for NF- κ B activation was similar between imatinib-insensitive DP cells and HSCs. Therefore, IL-1R/TLRs are likely responsible for maintenance of NF- κ B activation in imatinib-sensitive DP cells. We have added expression data of Tnf and other relevant receptors on NF- κ B activation in new Figures S2A and S2B and amended the main text in lines 186-191.

Wolf, A.M., Wolf, D., Rumpold, H., Ludwiczek, S., Enrich, B., Gastl, G., Weiss, G., Tilg, H., 2005. The kinase inhibitor imatinib mesylate inhibits TNF-production in vitro and prevents TNF-dependent acute hepatic inflammation. *Proceedings of the National Academy of Sciences* 102, 13622–13627. <https://doi.org/10.1073/pnas.0501758102>

8. The concept of targeting IRAK1/4 in hematological malignancies is not entirely novel, e.g.: PMID 29743719 and 32395555. This must be discussed. Thanks for your comment. We have added the sentences in discussion in lines 343-351.

9. It is important to consider that any in vivo treatment will also target the bone marrow microenvironment which is likely contributing to the inflammatory state. This should be carefully discussed. Thank you for your valuable suggestion. We have added a new paragraph in discussion in lines 369-381.

10. Figure S1C: While this reviewer agrees that the cells in this figure (SP and DP) look like blasts, there are clear differences in the size. Also, CML stem cells are not usually described as blasts (and neither are CML cells in chronic phase). Why are the cells in the G0+/CD27- more differentiated than the cells in

DN? That seems hard to understand. There is also a discrepancy between text and figure legend about the staining. Surely, these are Giemsa stainings (and not H&E)?

Thank you for your critical questions. The different sizes of DP and SP cells may be due to cell cycle status. Non-cycling cells should be smaller than cycling cells, especially M-phase cells. About the blast-like morphology, Kobayashi et al also showed the similar morphology of CML-LSCs (Kobayashi et al., 2014). The discrepancy may be due to the different disease state, because mouse CML-like disease is between chronic and blast phase in human CML, considering its rapid progression. The morphology of single G0+ positive cells are like those of mast cells. The surface phenotype of mast cells is similar as KSL cells. That is why they are in the KSL fraction. Kobayashi et al also reported the similar mast-like cells in the KSL fraction, which did not have leukemia-initiating ability. Finally, as for the staining method, we used hemacolor (Merck) which gives us similar staining results as Giemsa staining. To clarify this, we have now added the description.

Kobayashi, C.I., Takubo, K., Kobayashi, H., Nakamura-Ishizu, A., Honda, H., Kataoka, K., Kumano, K., Akiyama, H., Sudo, T., Kurokawa, M., Suda, T., 2014. The IL-2/CD25 axis maintains distinct subsets of chronic myeloid leukemia-initiating cells. *Blood* 123, 2540–2549. <https://doi.org/10.1182/blood-2013-07-517847>

Minor:

At the end of the sentence in the legends the authors frequently write 'combination of', but it should just say 'combination'.

Thank you. We have amended.

Reviewer #3 (Remarks to the Author); expert on clinical CML:

The authors present interesting data on the potential of IRAK1/4 inhibitors to

eradicate early CML progenitor cells. In addition to the cell line and patient cell data, and the animal studies presented, they add the anecdotal evidence from a patient who received dasatinib for CML in addition to nivolumab for a concomitant diagnosis of renal cell cancer. They report an early rapid deep molecular response that the authors suggest may be due to the combination therapy. Both the preclinical and the clinical data are of interest, but warrant some comments:

1. A deep molecular response can hardly be interpreted as eradication of the leukemic stem cells, the presumed benefit of the combination being used. Proof of this is that at least half of patients with sustained deep molecular responses must resume therapy after treatment discontinuation because of loss of major molecular response and yet others have detectable disease albeit at lower levels. Thus, the case being presented is not supportive of the main mechanism being proposed for this combination.

Thank you for your valuable opinion. We completely agree with your opinion. A deep molecular response does not mean effective eradication of CML LSCs. To examine this, the long observation of more cases must be needed. (Blast MRD CML 1 Trial is now on going. We hope that the trial will support our case.) In response to the reviewer's comment, we have weakened the discussion in lines 403-405.

2. Undetectable disease cannot be evaluated unless the control gene is known and the number of copies of it are available for the sample in question. The authors should provide both.

Thank you very much for raising the important point. The copy numbers of *BCR-ABL* and *ABL*(control) are as follows; 7.16 and 263921(6 month), 3.05 and 244113(9 month) and 0.00 and 135146(12 month). Therefore, this test is guaranteed. We have now provided it in the figure legend for Figure5 in lines 689-692.

3. The response of the patient was indeed rapid, but not unheard of. Some studies (e.g., Maiti et al. Cancer 2020; Naqvi et al., Cancer 2020) have reported

MR4.5 in approximately 20% of patients by 6 months, making this one instance not at all outside of the confidence interval of possibilities.

Thank you for your valuable comment. We agree with you. Because we have shown only one case as an example, we did not attempt to insist the efficacy of the combination therapy based on results of a single case. We have further weakened our statement in the revised version to avoid any confusion, quoting to the two studies you indicated in lines 403-405.

4. All the preclinical data presented relates to imatinib, whereas the patient in question is treated with dasatinib. One cannot assume the effects can be fully extrapolated. Dasatinib for example has been reported by some to have immunosuppressive effects that may make the immune properties of this combination different from those of imatinib.

Thank you for presenting such a valuable question. As the reviewer indicated, imatinib and dasatinib have different effects on immunological response. We have added some description in the discussion in lines 407-410. However, in this work, we attempted to show the efficacy of the combination between imatinib and PD-L1 antibody or IRAKi and did not intend to compare the effect between imatinib and dasatinib, which would be done a different work. (As this discrepancy is interesting, we will examine this in the future study.)

5. Trials using the combination of checkpoint inhibitors and TKIs have been conducted (e.g., NCT02011945). This should at least be acknowledged even of data is not available at this time.

Thank you for your comment. We have cited this trial (<https://doi.org/10.1182/blood-2020-137734>) in the discussion in lines 410-412.

REVIEWER COMMENTS

Reviewer #1 (Remarks to the Author):

The authors adequately addressed my comments and thereby improved the manuscript.

I have only one minor comment:

The newly included statement "Since PD-L1 induces antibody-dependent cell-mediated cytotoxicity (ADCC), we also assessed PD-1 and CTLA4 which do not induce ADCC." (page 13) is not correct. The ability of an antibody to mediate ADCC is not dependent on the recognized antigen, but rather the isotype of the antibody. Antibodies used in mouse studies that are specific for PD-L1 often kill tumor cells as they express PD-L1. To really prove that the treatment effect with the antibodies is due to immune checkpoint blockade and not to ADCC, PD-1 or CTLA4-specific antibodies are more suitable, as tumor cells usually do not express these two receptors.

Reviewer #2 (Remarks to the Author):

In this revised version of the manuscript by Tanaka et al. the authors, overall, have done a good job at addressing the reviewers' concerns. Just a few questions remain:

1. Are the mice in the secondary transplantation experiments really dying of CML, which is usually not the case in this murine model? What characterizes the mice at the time of death?
2. There is a misconception in the rebuttal letter about the nature of the disease in this model of CML. While the disease 'clinically' resembles that of an acute leukemia (death usually within 25 days), the cells morphologically are all mature neutrophils, i.e. resembling the chronic phase of the disease. Blasts are not found.
3. There continue to be multiple issues with English syntax and grammar.
4. On page 12, lines 252-253 reference is made to Figure S4E, although the text seems to be referring to Figures S3E+F.
5. Page 15, line 316: It would be best to mention CML stem cells here instead of CML reconstitution activity.
6. Discussion, page 18, line 387: Why do the authors claim that IRAK inhibitors would only decrease inflammation in the endosteal niche? It would reduce inflammation in the entire niche. The very formal separation into an endosteal and a vascular niche is no longer really made.
7. Figure 6B: How are the LSK cells defined?

Reviewer #3 (Remarks to the Author):

The authors have addressed my comments. I have no additional observations.

REVIEWER #1

The authors adequately addressed my comments and thereby improved the manuscript.

I have only one minor comment:

The newly included statement “Since PD-L1 induces antibody-dependent cell-mediated cytotoxicity (ADCC), we also assessed PD-1 and CTLA4 which do not induce ADCC.” (page 13) is not correct. The ability of an antibody to mediate ADCC is not dependent on the recognized antigen, but rather the isotype of the antibody. Antibodies used in mouse studies that are specific for PD-L1 often kill tumor cells as they express PD-L1. To really prove that the treatment effect with the antibodies is due to immune checkpoint blockade and not to ADCC, PD-1 or CTLA4-specific antibodies are more suitable, as tumor cells usually do not express these two receptors.

Thank you for pointing out our misunderstanding. According to your comment, we have amended the sentence as follow; “To prove that the treatment effect with the anti-PD-L1 antibody was due to immune checkpoint blockade and not to antibody-dependent cell-mediated cytotoxicity (ADCC), we also assessed PD-1 and CTLA4 which do not induce ADCC.”

REVIEWER #2

In this revised version of the manuscript by Tanaka et al. the authors, overall, have done a good job at addressing the reviewers' concerns. Just a few questions remain:

1. Are the mice in the secondary transplantation experiments really dying of CML, which is usually not the case in this murine model? What characterizes the mice at the time of death?

Thank you for the reviewer's comment. One could easily establish the secondary and tertiary transplantation experiments in this mouse CML-like model which requires some skills and experience. We have confirmed the increased GFP chimerism and splenomegaly in the moribund CML mice.

2. There is a misconception in the rebuttal letter about the nature of the disease in this model of CML. While the disease 'clinically' resembles that of an acute leukemia (death usually within 25 days), the cells morphologically are all mature neutrophils, i.e. resembling the chronic phase of the disease. Blasts are not found.

We agree with the reviewer. The comment "mouse CML-like disease is between chronic and blast phase in human CML, considering its rapid progression." was not appropriate. We simply meant that progression was quicker than human CML. We very much appreciate the reviewer for correcting our description in the last rebuttal letter. As reviewer indicated, CML cell consists of mostly differentiated myeloid cells including neutrophils.

3. There continue to be multiple issues with English syntax and grammar. Thank you for your comments. We tried to recheck and amend them.

4. On page 12, lines 252-253 reference is made to Figure S4E, although the text seems to be referring to Figures S3E+F. Thank you. We have amended it.

5. Page 15, line 316: It would be best to mention CML stem cells here instead of CML reconstitution activity. Thank you for your suggestion. According to the reviewer's suggestion, we have now added the following sentence at the end of the paragraph: "Collectively, the inhibition of IRAK1/4-NF-kB-PD-L1 pathway effectively decreases CML LSCs".

6. Discussion, page 18, line 387: Why do the authors claim that IRAK inhibitors would only decrease inflammation in the endosteal niche? It would reduce inflammation in the entire niche. The very formal separation into an endosteal and a vascular niche is no longer really made. Thank you for raising the important suggestion. Recent study showed that the endosteal niche is critical for the maintenance of CML LSCs. That's why we have just focused on the endosteal niche. As you mentioned, however, IRAK

inhibitors must affect the entire niche. Thus, according to your comment, we used BM leukemic niche instead of the endosteal niche.

7. Figure 6B: How are the LSK cells defined?

Thanks. Here, LSCs means DP cells. We estimated the absolute number of DP cells from the proportion of DP cells in the rest of whole bone marrow.

REVIEWER #3

The authors have addressed my comments. I have no additional observations. Thank you very much for your observations.